# Evaluation of Physical Egg Quality Parameters of Commercial Brown Laying Hens Housed in Five Production Systems

**DOI:** 10.3390/ani13040716

**Published:** 2023-02-17

**Authors:** Benjamin N. Alig, Ramon D. Malheiros, Kenneth E. Anderson

**Affiliations:** Prestage Department of Poultry Science, College of Agriculture and Life Sciences, North Carolina State University, Raleigh, NC 27606, USA

**Keywords:** egg quality, brown egg layers, Haugh unit, shell color, free-range, cage-free, enrichments

## Abstract

**Simple Summary:**

Avian welfare has become a highly debated issue for many activist groups and legislators over the past several years. Recently, the EU and several US states have banned the use of traditional cage environments. Therefore, it is important for producers and researchers to understand how this shift toward extensive environments will affect the quality of eggs being produced. The objective of this study was to identify how the five most used environments (cages, barren colony cages, enriched colony cages, cage-free, and free-range) affects the egg quality of brown egg layers (shell color, Haugh unit, yolk color, vitelline membrane, and shell strength). This study found that free-range eggs had superior egg quality parameters while, in most cases, eggs from both colony cages had inferior quality parameters. It was also found that both barren and enriched colony cages were not different in production parameters, indicating that simply adding enrichments to a cage environment will not change egg quality. This information shows that, as the egg industry moves toward greater adoption of extensive environments, egg quality will improve for brown egg layers, possibly allowing some of the costs to be offset.

**Abstract:**

This study evaluates the effect of housing environment on the egg quality characteristics of brown egg layers as many different environments are currently used in the industry. Battery cages, barren colony cages, enriched colony cages, cage-free, and free-range environments were evaluated. Overall, all egg quality measurements were affected by housing environment (*p* < 0.01) except for vitelline membrane strength, elasticity, and egg solids. Eggshells and yolks were lightest in barren colony cages and darkest from free-range hens (*p* < 0.0001). Free-range eggs were heavier than eggs from all other environments (*p* < 0.0001). Cage-free eggs had lower albumen height and Haugh units than other environments (*p* < 0.0001). Lastly, cage-free and free-range eggs had stronger eggshells than the other environments (*p* < 0.0001), and free-range eggs had more elastic eggshells than eggs from conventional battery cages and barren colony cages (*p* < 0.01). Access to the range seemed to give free-range hens different nutritional advantages, which allowed for the darker yolks and shells. Furthermore, eggs from barren colony cages seemed to exhibit more negative characteristics. Simply adding enrichments to colony cages did not improve or detract from egg quality. From this research, it appears that, as the industry moves toward extensive environments, the egg quality of brown egg layers will improve.

## 1. Introduction

Eggs are a staple of many diets around the world because many consumers believe eggs to be an economical source of calories and animal protein [1]. In the United States, most eggs are produced in conventional battery cage environments. According to the United Egg Producers, nearly 71% of all eggs produced in 2021 were produced in conventional cage systems [2]. However, the number of cage-free hens has been increasing steadily over the past several years, with around 50 million more cage-free hens reported to be part of the national flock in April 2022 compared to April 2019 [3,4,5,6]. Furthermore, the EU does not utilize barren colony cages or conventional battery cages, specifically relying on enriched colony cages and the cage-free environment for egg production [7]. This increasing trend is being driven by both legislation and food suppliers, which in turn are being driven by special interest and lobbying groups for animal rights and welfare such as the Humane Society, which began this trend by lobbing heavily for California’s ban on caged layers [8]. There are currently nine states in the US that have both banned the practice of battery cages for commercial layers and banned the sale of eggs from caged layers within the state [9]. Furthermore, with pressure from legislations and pressure from special interest groups, many grocery stores and food services are planning to end the sale of eggs from conventional cages and replace these eggs with eggs from alternative cages [10,11,12,13,14,15,16]. Unfortunately for egg producers, moving toward extensive environments requires substantially more manhours to manage and, therefore, greater expenses [17]. Studies have shown that consumers prefer eggs from extensive environments. However, purchases indicate they may not be willing to pay the increased price associated with the extensive environments [18,19]. However, consumers may not have a choice due to many external pressures on the industry [20].

Physical egg quality parameters, such as shell and vitelline membrane strength, shell color, Haugh unit, yolk color, and dry egg mass, can affect consumer perceptions, as well as industry product specifications and the uniformity of both table eggs and broken-out egg products [21,22]. Shell strength is important for table eggs, as eggs with weak shells have a much higher chance of breaking in transportation, while they are also important in the breaker market since strong shells break better. The vitelline membrane holds the yolk together and is, therefore, important in both table eggs (as some consumers prefer their yolks not broken during cooking) as well as keeping the yolk separate from the albumen when eggs are further processed [22,23]. Egg weight correlates to both the marketable egg size and the amount of product for further processing eggs. Haugh unit and albumen height speak to internal egg quality. Shell and yolk color are both important for consumer perceptions, as many US and EU consumers believe that darker egg yolks and shells are preferable. Lastly, egg solids are important for further processed eggs, particularly dry egg products [24,25,26]. In general, it has been shown that egg quality can be significantly influenced by housing environment, as many researchers theorize that the change in enrichments, movement space, or access to a range (with further access to sunlight, forage, insects, and more room for exercise) or wood shavings can lead to differences. Therefore, it is important to evaluate these changes [27,28,29,30,31,32].

Several studies have shown that extensive housing environments have a significant effect on egg quality parameters. While research indicates that allowing hens more freedom in their environments does not cause a significant increase in egg defects, it appears that physical quality characteristics are impacted [33]. Many studies agree that free-range environments and cage environments result in the heaviest eggs, while cage-free eggs are generally lighter [27,28,29,30]. Some researchers established that Haugh unit was better in free-range hens [30,34], while others showed that Haugh unit is superior in cage hen eggs [32,35]. Some researchers also identified that free-range hens had stronger eggshells [34] but other research identified that hens from conventional cages laid stronger eggshells [27,35]. Furthermore, some research concluded that hens in cages produce darker yolks than cage-free hens, while other papers identified that the cage-free system produces darker yolks [30,36,37,38]. Concerning shell color, some studies identified darker eggshells in cage-free environments [30]. Interestingly, there also appears to be a lack of research on the vitelline membrane strength and elasticity, as well as dry whole-egg matter. However, several studies have also been published demonstrating no effect of housing environment on egg quality [28,39,40,41,42]. Looking at previous research, there appears to be conflicting results. Furthermore, no one study simultaneously evaluated all the most common laying environments. Additionally, a USDA blueprint for animal research identified a need for replicating studies across different geographical locations to avoid confounding factors due to nutrition, management, or environment [43]. Many of the studies that evaluated housing environment were performed in other regions outside of the USA with different nutrition and management practices. Our objective for this study was to evaluate the physical egg quality of brown egg layers across differing production systems utilizing genetic strains, nutritional formulations, and management practices common to North America. While research is conflicting, many studies listed above showed a positive effect of the free-range environment on egg quality; therefore, we hypothesize that free-range brown egg layers have superior egg quality, that caged brown egg layers have poor egg quality, and that enrichments positively affect egg quality.

## 2. Materials and Methods

This trial was performed in conjunction with the 40th layer performance test (NCLP&MT) at the North Carolina Department of Agriculture and Consumer Services Piedmont Research Station Poultry Unit [44]. Two brown strains were utilized; however, strain effects are not identified in tables or figures. These strains were Lohman LB Lites (Lohmann Tierzucht GmbH, Cuxhaven, Germany) and Hy-Line Brown (Hy-Line International, West Des Moines, IA, USA), with 824 hens for each strain, totaling 1648 hens utilized for the study. All of the pullets were grown in accordance with the 40th NCLP&MT in the same environment in which they would be producing eggs [45]. At the end of the 16th week of age, pullets in the brood/grow house were moved to conventional cages, enrichable barren colony cages, or enriched colony cages.

### 2.1. Housing Environments

Hens were divided into five different housing environments: conventional cages (CC), barren enrichable colony cages (CS), enriched colony cages (ECS), cage-free (CF), and free-range (FR). The conventional cage laying system was designed as a standard closed-sided house with forced ventilation. The cages were arranged in three tiers and utilized manure belts under each tier. Each cage measured 40.6 cm (16 in) high by 50.8 cm (20 in) deep by 121.9 cm (48 in) wide, thus providing 6131.6 cm^2^ (6.66 ft^2^) for 12 hens per cage with a stocking density of 516 cm^2^ (80 in^2^) per bird. There were a total of 12 replicate units of two cages of 12 hens (24 hens per replicate) used for this trial. Each replicate unit of birds was fed by a trough system located on the outside of the cage. Nipple drinkers in each cage provided the birds with water.

A single windowless, force-ventilated house contained 18 replicate cages of each of the two types of colony cages. This house contained three-tiered banks of either enriched or enrichable colony cages. This house utilized a system that allowed for control of feed (amount and diet) to each replicate, and each cage was equipped with nipple waterers. There was no difference in size between the enriched cages and the enrichable colony cages. Each cage was 53.3 cm (21 in) tall by 66 cm (26 in) deep by 243.8 cm (96 in) wide, thus providing 1.6 m^2^ (17.3 ft^2^) for 31 birds per cage at a stocking density of 516 cm^2^ (80 in^2^) of cage space per bird. The major differences between the enriched and enrichable cages were the contents of the cages. The enrichable cages were barren colony cages that contained only the feeder and waterer systems. In contrast, the enriched cages contained several environmental enrichments, including nest boxes, roosts, and a scratch area.

The environmentally controlled cage-free house was a high-rise design with slatted flooring and manure pit beneath. Each 2.43 m × 3.05 m (8 ft × 10 ft) pen had 7.4 m^2^ (80 ft^2^) of floor space, half of which was covered in shavings and half of which was convered in slats. A total of ten replicates of cage-free birds were used for this study. For each replicate, 65 chicks were placed in each pen at a density of 1141 cm^2^ (177 in^2^) per bird. Feeders and waterers were placed in accordance with United Egg Producers’ guidelines. When the laying period began at 17 weeks of age, the replicate population was adjusted to 60 birds per pen with a flock density of 1233 cm^2^ (192 in^2^) per bird. After subtracting the space utilized for the feeders, the area per hen was 1141 cm^2^ (177 in^2^). The birds were provided with feeder and waterer space in accordance with UEP guidelines. Each hen was provided 16 cm of roosting space, and each pen contained 12 nesting boxes for a total of one nest box per five hens.

The free-range houses were curtain-sided houses with slatted floors and an attached outdoor section (paddock), which was completely enclosed with wire and included a net covering the top. The slats allowed the waste to drop below the house for easy removal without disturbing the birds. Once the hens were fully feathered, the heaters turned on only when the temperature dropped below 7.2 °C (45 °F) to keep the birds in their effective thermal neutral zone. Supplemental lighting was provided for the birds to match the lighting schedules of the other houses. At 12 weeks of age, the pullets were allowed access to their paddock. Both the pullets and the layers were fed ad libitum. Feed and water were provided for the birds inside the coop and in the paddock. Each replicate unit had eight nipple drinkers inside the coop and eight nipple drinkers outside in the paddock. Tube feeders were placed inside the coop, and a covered feeder was placed outside in the paddock. Both feeders combined allowed for 6.4 cm of feeder space per bird. Each free-range replica had identical dimensions. The 4 m × 2 m (12.1 ft × 6.6 ft) house portion of each replica measured 7.4 m^2^ (80 ft^2^) with 60 birds per pen, which yielded a house stocking density of 1141 cm^2^ (177 in^2^) per bird. Each paddock was measured at 18.3 m × 18.3 m (60 ft × 60 ft). To preserve forage quality, a diagonal fence split each paddock exactly in half, and the hens were rotated into a different section every 4 weeks. One week before the paddock rotation, the grass inside the unused pen was mowed to a height of 15 cm (6 in). The fences of the paddock were 1.8 m (6 ft) high with mesh covering the top to prevent birds from escaping and predators from entering. This rotating paddock design allowed for a stocking density of 2.8 m^2^ (30 ft^2^) per bird in the paddock.

### 2.2. Feeding and Lighting Program

The feeding program was designed as a phase feeding program to meet or exceed all nutrition requirements according to standard industry practices and National Research Council recommendations [46]. Table 1 shows that the diets that were fed on the basis of production and feed consumption; Table 2 shows what diets were fed on the basis of feed consumption and egg production. Furthermore, the lighting program followed a standard industry lighting schedule.

### 2.3. Egg Quality Measurements

All egg quality measurements were conducted by laboratory personnel who were trained to operate all egg quality equipment. All egg quality parameters were performed at 27, 35, 51, 63, 75, and 87 weeks of age unless otherwise noted. Egg quality was conducted using six eggs per replicate. The quality parameters measured were shell strength, shell elasticity, vitelline membrane elasticity, vitelline membrane hardness, egg weight, albumen height, Haugh unit (HU), yolk color, egg solid percentage, and shell color. Shell strength and shell elasticity were determined using a texture analyzer (TA-HDplus) with a 250-load cell measuring in grams of force. Vitelline membrane strength was determined using the TA.XTplus Texture Analyzer (Stable Micro Systems, Surrey, UK) with a 1 mm blunt probe with a 5 kg load cell per the manufacturer’s instructions. Haugh unit and albumen height were analyzed using the TSS QCD System (Technical Services and Supplies, Dunnington, York, UK) [47]. HU was calculated using the following equation [47]:Haugh Unit = 100 × Log(albumen height − 1.7 × egg weight + 7.6).

Yolk color was also determined using the TSS QCD System yolk color scan. Yolk color scan was calibrated using the DSM Yolk Color Fan that determines the color density from lightest to darkest with a range of 1–15 [48]. Shell color was determined using refractometry of black, blue, and red wavelengths combined to provide a reflectance score from 83.3% (white) to 0% (black). Whole-egg solid analysis was performed only at weeks 35, 63, and 75. All six eggs from each cage were combined and mixed in a stomacher for 30 s. While eggs were being mixed, a metal pan was weighed and recorded. After mixing, the metal pan was filled with egg mixture, weighed, recorded, and placed in a drying oven until dry at 50 °C. The dry matter in the pans was taken out, weighed, and recorded. The solid percentage was calculated using the following formula:Egg solid % = ((dw − pw) × 100)/(ps − pw),
where dw is the dry sample and pan weight, pw is the pan weight without sample, and ps is the pan and liquid sample weight.

### 2.4. Statistical Analysis

Data were analyzed utilizing JMP 15.2 using ANOVA, and treatments were determined to be statistically different from one another using Tukey’s HSD test [49]. Housing system, period, and their interaction were the main effects. There were significant period effects; therefore, data by period are presented as means in all tables. Strain effects were found to be significant; however, they were not found to impact the significance of treatment and, therefore, were not included in the model. As a note, in Section 3, treatment effects described as better, worse, higher, or lower are assumed statistically significant (*p* ≤ 0.05). Each housing environment treatment had a total of eight replicates for a total of 40 replicates.

## 3. Results

### 3.1. Shell Color

Housing environment had a significant effect on shell color overall, as well as during weeks 35, 51, and 63, as presented in Table 3. During week 35, hens in both colony cage environments were shown to have the lightest eggs compared to the CF and FR hens. The eggs from CC hens were not significantly different from CS, ECS, or CF; however, the CC eggs were shown to be lighter than the FR eggs. During week 51, CS hens had lighter-colored eggs than ECS, CF, and FR hens. Furthermore, while CC eggs were not statistically differently colored than CS, ECS, or FR hens, eggs from the CC environment were shown to be lighter-colored than eggs from the CF environment. Week 63 followed the same trend as week 35. Eggs from both colony cages were found to be lighter than eggs from the CF and FR hens, while eggs from the CC environment were also statistically lighter in color than only the FR eggs. Overall, CS eggs were statistically lighter in color than eggs from CC, CF, and FR environments but not statistically different from ECS eggs. Furthermore, FR eggs were darker in color than both CS and ECS eggs but were not statistically different in color compared to eggs from the CF and CC environments. Over time, for all environments, shell color started off darker and became slightly lighter in color, before remaining this color, although FR eggs seemed to oscillate between lighter and darker eggs each week of measurement.

### 3.2. Egg Weight

Table 4 shows the average eggs weight in grams. A statistically significant difference was established for weeks 27, 35, 51, and 63, as well as overall. Across each significant week and overall, hens in the free-range environment laid heavier eggs than all other environments, while hens in the other environments were not statistically different from each other. Over time, eggs in all environments increased in mass as the hens aged.

### 3.3. Inner Thick Albumen Height

The data in Table 5 denote an inner thick albumen height for the study. We did not find any statistical differences during any specific weeks; however, there was an overall significant difference with CF eggs showing lower albumen height by approximately 1 mm compared to the other environments. There were no differences among CC, CS, ECS, and FR eggs in terms of inner thick albumen height. As the hens aged, albumen height also slightly decreased for each environment.

### 3.4. Haugh Unit

Haugh unit, shown in Table 6, is a measurement that is determined by both egg weight and albumen height. We discovered no significant difference in Haugh unit during each individual sampling week, although the Haugh unit appeared to decrease throughout the life of the hen. Overall, we found that CF hens had lower Haugh unit by 5 than the other environments. CC, CS, ECS, and FR eggs did not show any differences in Haugh unit.

### 3.5. Yolk Color

The effect of housing environment on yolk color is presented in Table 7. Higher numbers denote darker yolks. As can be seen, the housing environment had a highly significant effect on yolk color every week and overall. There did not appear to be any trends in yolk color as hens aged for any environment. During each sample week, the free-range hens had darker yolks than other environments (except for cage-free during weeks 75 and 87) by 1.10–1.33 for week 27, 1.20–2.12 for week 35, 1.36–1.92 for week 51, 1.2–2.12 for week 63, 0.71–1.14 for week 75, and 1.02–1.52 for week 87. Furthermore, CF hens also had darker egg yolks than CS and ECS hens during weeks 27, 35, and 63 by approximately 0.50, 0.80, and 0.80, respectively. CF hens also had darker yolks than CC and CS during week 75 by approximately 0.60. Lastly, CF and CC hens had darker yolks than ECS hens during week 51 by 0.49. There were no statistical differences between the colony cage environments on any sampling dates. Overall, following similar trends to each sampling date, the FR hens had the darkest yolks by 0.92–1.58 compared to all other environments. CF hens also had darker yolks than CC, CS, and ECS hens by 0.41, 0.66, and 0.57, respectively. Lastly, CC hens had darker egg yolks than CS hens by 0.25. There were no differences between colony cage environments.

### 3.6. Eggshell Strength and Elasticity

Housing environment also had a significant effect on eggshell strength and eggshell elasticity as can be seen in Table 8 and Table 9. In general, shell strength trended downward as the hen aged for each environment. Housing environment was shown to have a significant effect on shell strength during all weeks (except for week 75) and overall. During week 27, FR hens had stronger eggs than CS and ECS eggs by 8.80 N/mm and 10.35 N/mm, respectively. The egg strength was not significantly different between CF and CC for week 27. A similar trend appeared during week 35 and week 51, where CF and CFR hens had stronger eggs than CC, CS, and ECS hens by 8.88–13.44 N/mm for week 35 and 9.06–12.75 N/mm for week 51. During week 63, FR hens had stronger eggs than CC, CS, and ECS hens by 10.05 N/mm, 8.82 N/mm, and 6.71 N/mm, respectively. Furthermore, CS hens had weaker eggs than CF hens by 8.15 N/mm. There was no difference in CC, CS, and ECS eggshell strength for week 63. During the final week, FR hens had stronger eggs than ECS hens by 9.05 N/mm. No other environments were significant during this week. Overall, FR and CF hens had stronger eggshells than CC, CS, and ECS hens by 4.94–9.18 N/mm.

Housing environment had a significant effect on shell elasticity during weeks 27 and 35, as well as overall. During week 27, FR eggs were more elastic than ECS eggs by 0.087 mm. CC, CS, and CF egg elasticity was not significant. During week 35, CF eggs were more elastic than CS and CC eggs by 0.097 mm and 0.103 mm, respectively. Furthermore, FR eggs were also found to be more elastic than CS eggs by 0.092 mm. ECS eggs were not significantly different from the other environments. Overall, FR eggs were more elastic than CC and CS eggs by 0.032 mm and 0.022 mm, respectively, and CF eggs had more elastic shells than CC eggs by 0.030 mm. ECS eggs were not significantly different when compared to other environments.

### 3.7. Vitelline Membrane Strength and Elasticity

Table 10 and Table 11 denote vitelline membrane strength and elasticity, respectively. Housing environment significantly affected vitelline membrane strength only during week 35. During this week, FR eggs had stronger vitelline membranes than CS and ECS eggs by 0.0036 N/mm and 0.0046 N/mm, respectively. There was no significant difference in vitelline membrane strength overall. Lastly, we found no significant difference between housing environments for vitelline membrane elasticity during any timepoint or overall.

### 3.8. Whole-Egg Solids

Table 12 shows the whole-egg solid percentage by period and overall, for each environment. There was no significant difference in egg solids across housing environments for any of the production periods measured or overall.

## 4. Discussion

As shown by the results, housing environment had a highly significant effect on most physical egg quality parameters. Shell color intensity is an important marketing measurement in brown eggs, and preference is determined by region, such as some markets preferring dark eggs and some preferring lighter brown eggs [50]. In the USA specifically, most consumers find darker-brown eggshells to be preferable [26]. Our study identified that hens from both colony cages laid the lightest eggs while CC, CF, and FR eggs were the darkest, although CC, ECS, and CF eggs were not statistically different. Overall, our study agrees with several studies such as Ahmmed et al. [51] and Sokolowicz et al. [29], who discovered no difference between CF and CC color. Our study also partially agrees with Dedousi et al. [52], who found that CF and FR hens had the same egg color intensity, while our study showed that ECS had lighter eggs than the extensive environments, which disagrees with Dedousi et al. Furthermore, Roll et al. [42] also did not identify a difference between ECS and CF brown egg color, which agrees with our study. While we did not find any differences in CC, CF, and FR egg color, Samiullah et al. [30] observed that CC eggs were the darkest, followed by FR eggs, with CF eggs being the lightest. It is well known that brown eggshell color lightens as the hen ages due to the decrease in the amount of protoporphyrin IX deposited on the eggshell [30,53]. While the mechanism of protoporphyrin IX is unclear, most researchers believe that it is synthesized and deposited in the shell gland [54]. Therefore, the change in pigmentation could be due to the amount of time the egg spends in the shell gland. It is also understood that, as brown egg layers age, their eggs become lighter, as seen in other studies; while no analyses were performed on this, the averages across all housing environments followed this trend in our study [50,54]. These researchers believed that egg size is related to pigmentation intensity; however, as seen by our data, in which free-range hens had the largest eggs, this theory may not be true. A further theory on eggshell pigmentation intensity has to do with physiological stress [55]. The correlation between high stress and low pigmentation was discovered by Mills et al. [55], who identified that hens kept in higher densities in cages had greater stress parameters than hens in lower densities in single cages; however, unfortunately, research characterizing the stress response, as well as eggshell color, across all the most popular housing systems is lacking [56,57]. Therefore, it can be hypothesized that lighter eggshells in colony cage environments are indicative of lower amounts of protoporphyrin IX and higher stress levels in these environments. Further research will need to be completed to substantiate this claim.

Housing environment also had a major effect on eggshell strength and eggshell elasticity. We discovered that CF and FR hens had greater shell breaking strength than the other environments. Furthermore, we also found that shell elasticity followed similar patterns where FR eggshells were more elastic than CS and CC shells, CF shells were more elastic than CC shells, and ECS shells were not statistically different. Shell strength followed the exact same trend as shell color. The environments that had darker eggs also had stronger shells. As mentioned before, this could be due to the amount of time that the egg spent in the shell gland, but more research would need to be performed to determine this. Our results agree with several studies such as Dedousi et al. [52], Samiullah et al. [30], and Krawczyk et al. [38]. They observed that FR eggs were stronger than eggs from cages. Samiullah et al. [30] also disclosed no difference in shell breaking strength between CF and FR eggs. Dedousi et al. [52] also reported that CF eggs were weaker than FR eggs, and Samiullah et al. [30] reported no difference in CF, FR, and CC eggs, which disagrees with our study. Dikmen et al. [34] revealed no difference in CC, ECS, and FR eggs, which partially agrees with the present study. Ketta et al. [39] compared ECS and CF eggs and identified no difference in shell strength between these environments. Sokolowicz et al. [29] also reported that CF hens had stronger eggs than FR hens, which also disagrees with our results. Dong et al. [41] found no difference between CC and FR; however, this study was performed with wildtype chickens indigenous to China. Interestingly, no studies reported the elasticity of the shell. While strength is important, elasticity allows the eggshell to slightly bend when pressure is applied before it breaks. It is currently unknown what causes CF and FR hens to have superior eggshell quality. While shell thickness was not measured, this could be an explanation as to why these environments had stronger eggs.

Moving into the egg, albumen height and Haugh unit followed the same trend for this study, which is not surprising due to Haugh unit using albumen height in its calculation. Overall, CF eggs had lower albumen height and lower Haugh unit scores than the other environments. Interestingly, at each sampling point, there was no difference in Haugh unit or albumen height. Several studies agree with ours, identifying that CF eggs had lower Haugh units than other environments or that the other environments were not different [28,29,37,38,41,51]. However, a handful of studies either completely disagree or partially agree with ours. Golden et al. [58] found that CC eggs had higher Haugh unit than CF eggs. Englmaierova et al. [35] and Popova et al. [32] disclosed that, as housing system became more extensive, Haugh unit decreased. While Samiullah et al. [30] reported that CF and FR were statistically different, this study also found that FR was different from CC, and that CF and CC were not different. Dikmen et al. [34] discovered that CC and ECS Haugh unit was not different; however, FR was different from CC and ECS eggs. Lewko et al. [36] identified no difference among CC, CF, and FR. Dedousi et al. [52] detected no differences in Haugh unit among ECS, CF, and FR eggs. Haugh unit is the conventional predictor of internal egg quality, as egg quality is determined on albumen height. As the egg ages, Haugh unit decreases due to the loss of moisture and the breakdown of proteins inside the albumen [59]. It has been shown that different dietary elements, as well as stressors, can affect Haugh unit [60,61]. Therefore, CF hens may be experiencing poor gut health/absorption or different stressors, perhaps due to their proximity with their own fecal matter or access to wood shavings. More research is required to answer this question, such as evaluating gut health and stress markers from the hens.

When analyzing yolk color, our study ascertained that, as housing environments became more extensive, the yolks became darker. Several previous studies confirm our findings that yolks darken when environments become more extensive [29,36,37]. There are also several studies that identified no difference in yolk color as a function of environment [28,41,42,51]. Furthermore, a handful of studies identified that yolks were darker in intensive environments [38,62]. Lastly, a handful of studies partially agree, such as Dedousi et al. [52] who found that FR had the darkest yolks, followed by ECS, with CF having lighter yolks. It is well known that, in most markets in the world, consumers prefer darker yolks as they associate darker eggs with healthiness [24,63,64]. The color of the yolk is directly related to the amount of carotenoids within the yolk [65]. It is also well known that nutritional factors such as marigold, red pepper meal, or any additive with high levels of carotenoids will increase the pigmentation of egg yolks [66,67]. The only environment that had access to different nutritional factors was the FR environment, which had access to the range paddocks. Therefore, we hypothesize that the reason for FR eggs having darker yolks was the access to high-fiber forage and insects from the range paddocks; several studies directly feeding forage to hens support this theory [68,69]. CF hens also had darker yolk colors than other cage environments. As discussed for Haugh unit, higher levels of stress can affect yolk color. From previous research, it appears that higher levels of blood corticosterone cause darker yolks; however, this study used direct fed corticosterone as the stressor, and the study only lasted for 10 days [70]. Furthermore, higher corticosterone levels in the hen’s blood correlates to corticosterone [71]. Therefore, we hypothesize that CF hens under higher levels of stress lay eggs that contain higher levels of corticosterone, and that yolk color and egg corticosterone levels are related.

Lastly, our study did not find that housing environment influenced overall vitelline membrane strength, vitelline membrane elasticity, or egg dry matter percentage. Interestingly, previous research on housing environments and egg quality omitted vitelline membrane analysis and whole-egg dry matter percentages; therefore, these measurements are somewhat novel to this study. The most recent study that evaluated vitelline membrane strength from 2012 highlighted no difference between caged and cage-free layers, which does agree with our study [58].

## 5. Conclusions

From this study, we found that extensive environments, specifically FR and CF, showed superior egg quality traits, such as yolk color, shell color, and shell breaking strength, according to customer perception. However, CF eggs did demonstrate some deficiencies in some areas, such as egg weight and Haugh units. Moreover, it was found that CC environments produce the worst-quality eggs according to consumer and industry preferences. Lastly, this study displayed no major differences in egg quality between CS and ECS environments. Therefore, we partially accept our hypothesis that extensive environments have a positive effect on egg quality, and that intensive environments have a negative effect on egg quality. Moreover, we reject that enrichments improve egg quality, as there was no difference between CS and ECS environments; instead, we submit that enrichments alone have no effect on egg quality. Our research found that removing the hens from cages seemed to be the bigger factor in improving egg quality as there were also minimal differences between the colony cages and the CC system. Therefore, while simply adding enrichments did not affect egg quality, we propose that factors associated with CF and FR environments, such as access to litter, different nesting material, or higher levels of movement and exercise, can possibly play a larger role in improving egg quality than access to enrichments. Further research is needed to identify why these housing environments are causing these effects. Informed by other research, we believe that physiological and behavioral stressors can cause differences in egg quality; therefore, stress parameters, such as plasma corticosterone, should be measured and correlated with egg quality. Furthermore, while the results of this study show that FR eggs demonstrated superior quality, the cause is not well understood. Further research is needed to establish how different aspects of the free-range environment, such as access to forage, sunlight, exercise, or climatic stimuli, affect the egg quality of commercial brown egg layers. As demand for eggs from extensive environments is growing, it is increasingly important to understand how this will affect the product being produced. The improved egg quality could potentially bring an economic benefit to partially offset the costs of the extensive environment; however, more research is required in order to substantiate this claim. From the results of this study, it appears that producers may not need to expect a change in egg quality when adding enrichments to a colony cage environment. Furthermore, producers can expect darker yolks, darker shells, and stronger eggshells as environments become extensive, which follows consumer preferences. Lastly, according to the results of our study, producers should be aware that switching to a cage-free environment may have a poor impact on Haugh unit.

## Figures and Tables

**Table 1 animals-13-00716-t001:** Feeding program of diets according to egg production rate and ad libitum consumption rate.

Rate of Production	Feed Consumption g/100 Birds/Day	Diet Fed
Pre-production	<9.52	Pre-lay
	<10.43	Pre-peak
Pre-peak and >90%	10.43–12.20	Layer 1
	>12.20	Layer 2
	<11.29	Layer 2
90–80%	11.29–12.20	Layer 3
	>12.20	Layer 4
	<11.29	Layer 4
70–80%	11.29–12.20	Layer 5
	>12.20	Layer 6
	<11.29	Layer 6
<70%	11.29–12.20	Layer 7 ^1^
	>2.20	Layer 7 ^1^

^1^ Layer 7 did not get used during this study.

**Table 2 animals-13-00716-t002:** Ingredient composition and calculated nutrient analysis of diets fed to all hens according to the feeding program described in Table 1.

Ingredients	Pre-Lay	Pre-Peak	Layer 1	Layer 2	Layer 3	Layer 4	Layer 5	Layer 6
	(%)	(%)	(%)	(%)	(%)	(%)	(%)	(%)
Corn	48.7	58.3	60.1	62.0	68.0	66.5	65.8	65.2
Soybean meal	35.2	28.2	26.7	25.3	25.0	22.0	20.9	18.9
Wheat midds	-	-	-	-	-	-	5.70	12.90
Fat (lard)	0.55	0.50	-	-	0.83	-	-	-
Soybean oil	2.540	1.290	1.810	1.250	0.095	-	-	-
Lysine 78.8%	-	-	-	-	-	0.11	0.005	-
DL-Methionine	0.170	0.150	0.120	0.100	0.095	0.078	0.062	0.057
Ground limestone	6.87	6.12	6.08	5.53	-	5.78	5.96	6.18
Course limestone	3.87	3.50	3.50	3.75	3.97	3.75	3.75	3.75
Bicarbonate	0.11	0.10	0.10	0.15	0.11	0.10	0.10	0.10
Phosphate mono/D	1.21	1.07	0.90	1.30	1.26	1.09	0.99	0.82
Salt	0.39	0.32	0.29	0.25	0.31	0.26	0.26	0.24
Vit. premix ^1^	0.05	0.05	0.05	0.05	0.05	0.05	0.05	0.05
Min. premix ^2^	0.05	0.05	0.05	0.05	0.05	0.05	0.05	0.05
HyD3 broiler (62.5 mg/lb)	-	-	0.025	-	-	-	-	-
Prop acid 50% dry	0.055	0.050	0.050	0.050	0.053	0.050	0.050	0.050
T-Premix	0.055	0.050	0.050	0.050	0.053	0.050	0.050	0.050
0.06% selenium premix ^3^	0.055	0.050	0.050	0.050	0.053	0.050	0.050	0.050
Choline Cl 60%	0.090	0.097	0.080	0.050	0.046	0.026	0.005	-
Avizyme	0.055	0.050	-	-	-	-	-	-
Ronozyme P-CT 540%	0.022	0.020	0.020	-	-	-	-	-

Calculated values								
Crude protein %	19.43	18.10	17.50	17.00	16.37	15.87	15.49	14.93
Calcium %	4.10	4.05	4.00	3.95	3.95	4.00	4.05	4.10
A. Phos. %	0.45	0.44	0.40	0.38	0.35	0.33	0.31	0.28
Total lysine %	1.10	1.00	0.96	0.91	0.87	0.91	0.80	0.75
Total sulfur amino acids %	0.8	0.74	0.69	0.66	0.63	0.60	0.58	0.56
ME kcal/kg	2926	2904	2860	2843	2843	2822	2800	2778

^1^ Vitamin premix supplied the following per kilogram of feed: vitamin A, 26,400 IU; cholecalciferol, 8000 IU; niacin, 220 mg; pantothenic acid, 44 mg; riboflavin, 26.4 mg; pyridoxine, 15.8 mg; menadione, 8 mg; folic acid, 4.4 mg; thiamin, 8 mg; biotin, 0.506 mg; vitamin B12, 0.08 mg; ethoxyquin, 200 mg. The vitamin E premix provided the necessary amount of vitamin E as DL-α-tocopheryl acetate. ^2^ Mineral premix supplied the following per kilogram of feed: 120 mg of Zn as ZnSO_4_H_2_O, 120 mg of Mn as MnSO_4_H_2_O, 80 mg of Fe as FeSO_4_H_2_O, 10 mg of Cu as CuSO_4_, 2.5 mg of I as Ca(IO_3_)_2_, and 1.0 mg of Co as CoSO_4_. ^3^ Selenium premix provided 0.3 ppm Se from sodium selenite.

**Table 3 animals-13-00716-t003:** The effect of housing environment on shell color of brown egg layers by week and overall ^1^.

Housing Environment	Week 27	Week 35	Week 51	Week 63	Week 75	Week 87	Overall
Conventional cages (CC)	23.1	26.8 ^AB^	27.1 ^AB^	26.8 ^AB^	27.4	28.9	26.7 ^BC^
Enrichable colony cages (CS)	26.2	30.3 ^A^	28.5 ^A^	30.3 ^A^	27.0	28.8	28.5 ^A^
Enriched colony cages (ECS)	23.3	29.5 ^A^	26.3 ^BC^	29.5 ^A^	29.1	29.1	27.8 ^AB^
Cage-free (CF)	23.7	25.6 ^BC^	24.6 ^C^	25.6 ^BC^	27.4	29.3	26.3 ^BC^
Free-range (FR)	24.8	22.8 ^C^	25.5 ^BC^	22.8 ^C^	27.5	30.6	25.5 ^C^
Pooled standard error	1.06	0.96	0.54	0.96	1.20	1.01	0.40
*p*-Value	0.177	0.0001	0.0001	0.0001	0.760	0.753	0.0001

^1^ Shell color based on reflectance with pure white having a reflectance of 83.3% and pure black having a reflectance of 0%. ^A, B, C^ Denotes significant differences utilizing Tukey’s HSD test for separation of means and alpha = 0.05.

**Table 4 animals-13-00716-t004:** The effect of housing environment on egg weight (grams) of brown egg layers by week and overall.

Housing Environment	Week 27	Week 35	Week 51	Week 63	Week 75	Week 87	Overall
Conventional cages (CC)	60.1 ^B^	62.0 ^B^	60.9 ^B^	62.0 ^B^	66.5	66.4	63.0 ^B^
Enrichable colony cages (CS)	59.9 ^B^	62.3 ^B^	62.7 ^B^	62.3 ^B^	64.9	65.2	62.9 ^B^
Enriched colony cages (ECS)	59.8 ^B^	62.8 ^B^	62.0 ^B^	62.8 ^B^	65.9	66.1	63.2 ^B^
Cage-free (CF)	59.9 ^B^	60.9 ^B^	61.2 ^B^	60.9 ^B^	65.6	66.2	62.5 ^B^
Free-range (FR)	62.2 ^A^	68.0 ^A^	65.8 ^A^	68.0 ^A^	67.5	68.5	66.7 ^A^
Pooled standard error	0.61	0.69	0.18	0.69	1.17	0.92	0.34
*p*-Value	0.049	0.0001	0.0001	0.0001	0.607	0.161	0.0001

^A, B^ Denotes significant differences utilizing Tukey’s HSD test for separation of means and alpha = 0.05.

**Table 5 animals-13-00716-t005:** The effect of housing environment on albumen height (mm) of brown egg layers by week and overall.

Housing Environment	Week 27	Week 35	Week 51	Week 63	Week 75	Week 87	Overall
Conventional cages (CC)	9.43	8.99	7.53	8.99	8.10	7.64	8.44 ^A^
Enrichable colony cages (CS)	9.38	8.72	7.39	8.72	7.78	7.54	8.24 ^A^
Enriched colony cages (ECS)	9.13	8.54	7.48	8.54	7.69	7.61	8.16 ^A^
Cage-free (CF)	9.29	8.14	7.08	8.14	6.97	6.87	7.58 ^B^
Free-range (FR)	9.40	9.30	8.01	9.30	7.56	7.81	8.56 ^A^
Pooled standard error	0.35	0.32	0.23	0.32	0.40	0.37	0.14
*p*-Value	0.131	0.122	0.265	0.122	0.3703	0.4579	0.0001

^A, B^ Denotes significant differences utilizing Tukey’s HSD test for separation of means and alpha = 0.05.

**Table 6 animals-13-00716-t006:** The effect of housing environment on Haugh Unit of brown egg layers by week and overall.

Housing Environment	Week 27	Week 35	Week 51	Week 63	Week 75	Week 87	Overall
Conventional cages (CC)	96.1	93.8	86.0	93.8	86.5	84.9	90.2 ^A^
Enrichable colony cages (CS)	96.0	92.1	84.1	92.1	85.9	84.4	89.1 ^A^
Enriched colony cages (ECS)	94.8	90.9	85.1	90.9	85.0	84.1	88.5 ^A^
Cage-free (CF)	89.9	89.2	82.9	89.2	80.1	78.9	85.0 ^B^
Free-range (FR)	95.5	93.8	87.2	93.8	84.1	85.2	89.9 ^A^
Pooled standard error	1.75	1.76	1.87	1.76	2.04	2.53	0.81
*p*-Value	0.916	0.303	0.528	0.303	0.215	0.392	0.0001

^A, B^ Denotes significant differences utilizing Tukey’s HSD for separation of means and alpha = 0.05.

**Table 7 animals-13-00716-t007:** The effect of housing environment on yolk color (Rouche) of brown egg layers by week and overall.

Housing Environment	Week 27	Week 35	Week 51	Week 63	Week 75	Week 87	Overall
Conventional cages (CC)	8.25 ^BC^	7.96 ^BC^	7.77 ^B^	7.96 ^BC^	7.90 ^C^	8.08 ^B^	7.99 ^C^
Enrichable colony cages (CS)	8.02 ^C^	7.46 ^C^	7.21^C^	7.46 ^C^	8.33 ^BC^	7.69 ^B^	7.74 ^D^
Enriched colony cages (ECS)	8.04 ^C^	7.60 ^C^	7.50 ^BC^	7.60 ^C^	8.06 ^C^	8.19 ^B^	7.83 ^CD^
Cage-free (CF)	8.56 ^B^	8.38 ^B^	7.77 ^B^	8.38 ^B^	8.69 ^AB^	8.60 ^AB^	8.40 ^B^
Free-range (FR)	9.35 ^A^	9.58 ^A^	9.13 ^A^	9.58 ^A^	9.04 ^A^	9.21 ^A^	9.32 ^A^
Pooled standard error	0.12	0.16	0.13	0.16	0.15	0.18	0.061
*p*-Value	0.0001	0.0001	0.0001	0.0001	0.0001	0.0001	0.0001

^A, B, C, D^ Denotes significant differences utilizing Tukey’s HSD test for separation of means and alpha = 0.05

**Table 8 animals-13-00716-t008:** The effect of housing environment on shell strength (N/mm^2^) of brown egg layers by week and overall.

Housing Environment	Week 27	Week 35	Week 51	Week 63	Week 75	Week 87	Overall
Conventional cages (CC)	47.25 ^AB^	41.72 ^B^	39.20 ^B^	33.99 ^BC^	47.11	41.19 ^AB^	41.75 ^B^
Enrichable colony cages (CS)	44.81 ^B^	43.28 ^B^	40.03 ^B^	35.22 ^C^	40.38	38.58 ^AB^	40.39 ^B^
Enriched colony cages (ECS)	43.62 ^B^	40.06 ^B^	36.89 ^B^	37.33 ^BC^	41.86	35.30 ^B^	39.18 ^B^
Cage-free (CF)	49.54 ^AB^	52.16 ^A^	49.64 ^A^	43.37 ^AB^	43.46	42.01 ^AB^	46.69 ^A^
Free-range (FR)	53.61 ^A^	53.50 ^A^	49.10 ^A^	44.04 ^A^	45.51	44.35 ^A^	48.35 ^A^
Pooled standard error	1.83	1.64	1.66	1.51	1.89	1.83	0.71
*p*-Value	0.004	0.0001	0.0001	0.0004	0.108	0.015	0.0001

^A, B, C^ Denotes significant differences within the column utilizing Tukey’s HSD test for separation of means and alpha = 0.05.

**Table 9 animals-13-00716-t009:** The effect of housing environment on eggshell elasticity (mm) of brown egg layers by week and overall.

Housing Environment	Week 27	Week 35	Week 51	Week 63	Week 75	Week 87	Overall
Conventional cages (CC)	0.619 ^AB^	0.577 ^BC^	0.218	0.207	0.206	0.274	0.350 ^C^
Enrichable colony cages (CS)	0.624 ^AB^	0.571 ^C^	0.247	0.225	0.194	0.287	0.360 ^BC^
Enriched colony cages (ECS)	0.578 ^B^	0.644A^BC^	0.233	0.238	0.213	0.285	0.364 ^ABC^
Cage-free (CF)	0.623 ^AB^	0.674 ^A^	0.232	0.229	0.239	0.281	0.380 ^AB^
Free-range (FR)	0.665 ^A^	0.663 ^AB^	0.235	0.232	0.215	0.287	0.382 ^A^
Pooled standard error	0.0146	0.0224	0.0062	0.0092	0.0171	0.0081	0.0058
*p*-Value	0.005	0.004	0.052	0.614	0.443	0.692	0.009

^A, B, C^ Denotes significant differences within the column utilizing Tukey’s HSD test for separation of means and alpha = 0.05.

**Table 10 animals-13-00716-t010:** The effect of housing environment on vitelline membrane strength (N/mm) of brown egg layers by week and overall.

Housing Environment	Week 27	Week 35	Week 51	Week 63	Week 75	Week 87	Overall
Conventional cages (CC)	0.023	0.020 ^AB^	0.018	0.023	0.020	0.021	0.021
Enrichable colony cages (CS)	0.023	0.019 ^B^	0.018	0.020	0.021	0.021	0.020
Enriched colony cages (ECS)	0.023	0.011 ^B^	0.019	0.019	0.020	0.021	0.020
Cage-free (CF)	0.023	0.020 ^AB^	0.018	0.019	0.021	0.019	0.020
Free-range (FR)	0.023	0.023 ^A^	0.018	0.020	0.020	0.020	0.021
Pooled standard error	0.0009	0.0008	0.0008	0.0010	0.0010	0.0012	0.0004
*p*-Value	0.987	0.006	0.951	0.596	0.628	0.796	0.657

^A, B^ Denotes significant differences within the column utilizing Tukey’s HSD test for separation of means and alpha = 0.05.

**Table 11 animals-13-00716-t011:** The effect of housing environment on vitelline membrane elasticity (mm) of brown egg layers by week and overall.

Housing Environment	Week 27	Week 35	Week 51	Week 63	Week 75	Week 87	Overall
Conventional cages (CC)	3.83	3.51	3.26	3.85	3.31	3.22	3.50
Enrichable colony cages (CS)	3.80	3.48	3.16	3.50	3.68	3.41	3.50
Enriched colony cages (ECS)	3.90	3.31	3.34	3.38	3.60	3.35	3.48
Cage-free (CF)	3.67	3.41	3.07	3.29	3.67	3.15	3.37
Free-range (FR)	3.92	3.89	3.04	3.16	3.16	3.09	3.38
Pooled standard error	0.175	0.216	0.18	0.179	0.194	0.236	0.081
*p*-Value	0.860	0.401	0.720	0.438	0.234	0.861	0.650

**Table 12 animals-13-00716-t012:** The effect of housing environment on whole-egg solid percentage of brown egg layers by week and overall.

Housing Environment	Week 39	Week 63	Week 75	Overall
Conventional cages (CC)	24.0	23.4	24.1	23.8
Enrichable colony cages (CS)	24.3	23.5	23.9	23.9
Enriched colony cages (ECS)	24.1	23.3	23.8	23.8
Cage-free (CF)	24.0	23.4	24.2	23.9
Free-range (FR)	24.6	23.7	23.8	24.0
Pooled standard error	0.28	0.17	0.19	0.13
*p*-Value	0.504	0.619	0.433	0.612

## Data Availability

Data are available upon request.

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
