# Peer review of "Evaluation of Physical Egg Quality Parameters of Commercial Brown Laying Hens Housed in Five Production Systems"

_animals, 2023, doi:10.3390/ani13040716_

Round 1

Author Response

Hello Reviewer 1,

Thank you for your feedback. As I have been asked to address each comment individual I will list the comment and my response below.

Line 47-50 on vitelline membrane: please include a reference

I have added 2 references that discuss why vitelline membrane strength us important for the industry

Line 110-113: mentions 3 strains used but only two cited.

I have fixed this error. In my master’s thesis I included 3 strains however, we removed one of the strains to publish as one was an experimental strain.

Line 114: don’t forget to cite the grow report!

Thank you for catching this! Report is now cited

Line 116: add “barren” to “enrichable colony cages

Added the descriptor

Line 124: superscript “2” in 80in2

I have added the superscript as well as ensured that this is corrected in all other cases.

Line 117 (Table 1): I recommend removing Layer 7 from the table since it was not used.

Thank you for the recommendation but I kept layer 7 in the table in case a reader wondered what happened if hens consumed more feed than 11.29 g/100 birds/day at less than 70% production

Line 174: do you mean table 1?

Line 175: do you mean table 2?

Changed the table names to correctly reference the tables. (I had previously had more tables in this manuscript but removed some)

Line 181 (Table 2): Please adjust the font for table 2 to Times New Roman.

Thank you for the catch! I have changed the font to match that of the manuscript template

Line 192-200: I recommend moving the Stats Analysis section to the last paragraph after egg quality.

The stats section has been moved to the end of the M&M

Line 245 (Table 3): please superscript the different letters. Same for all other tables.

I have superscripted my multiple comparisons reports

Line 278: You had subheadings for each results section, however stopped at yolk color. I recommend choosing one system and being consistent with it.

Thank you for catching this! Section headings added.

Line 312: remove break into new paragraph

Removed

Line 417-418: Add references

References are on the next line.

Line 424-426 and 443-451: you mentioned CF hens may be under stress – what is the reason for this? Just speculating here, but could it have been that access to shavings and foraging altered yolk colour (just like access to foraging in FR)?

Thank you for the suggestion! Added a part with a reasoning for cage free hens having access to their own fecal matter.

Again thank you for your suggestions and I hope you find my edits satisfactory,

Benjamin Alig, Ramon Malheiros and Kenneth Anderson

Reviewer 2 Report

General comments:

The manuscript animals-2178064, entitled "The effect of housing environment on physical egg quality of commercial brown egg layers" with Mr. Alig as first author addresses the question of what impact the increasingly cage-free housing of laying hens might have on production parameters. The manuscript is well structured, carefully formulated and presents interesting ideas. However, some sections need to be revised and major mistakes must be corrected especially in the “material and methods” and “results” section. Various comments for changes to the text could be find in the following section.

Detailed comments:

L10: Please also provide data from other states e.g. from Europe.

L13: The connection to egg quality is not clear from the explanations. The beginning of the sentence with "Therefore" does not fit. Please rephrase this section.

L15: Please delete the first “and” in the brackets.

L25: Why are only brown egg layers considered? Are significantly different results to be expected for white egg layers? What is the underlying hypothesis?

L26: In my opinion it is not necessary to divide the abstract in smaller sections. Please delete the subheadings (methods, results, conclusion).

L26: Please justify the hypothesis why egg quality should improve in the free range environment.

L28: The abbreviations have not been introduced before. Please use the written words (UFP, NRC, HSD).

L32: You have used the word “found” very often. Please use other terms for variety.

L36: As in the simple summary, I would write one more sentence here on the practical implications. Why is this study important for the farms?

L38: Please delete the semicolon at the end of the keywords.

L44: I would refer here not only to consumer preferences, but also to industry requirements.

L54: Please give a reference for the statements concerning egg quality.

L54: Please add another sentence as a transition between the sections: e.g. Egg quality can be significantly influenced by the housing environment (add reference)

L55: What is the situation in other countries? Please insert information for e.g. Europe.

L68: What is meant by the term "cage-free"? Please define this term.

L70: This sentence is too long and complicated. Please split the sentence in two parts.

L73: This sentence is a repetition and should therefore be deleted.

L78: This sentence appears again below in a similar form and should therefore be deleted here.

L78: Please mention some influencing factors in this section. What could be the reasons for better or worse egg qualities (e.g. movement, light, nutrition...)?

L85: You have used the word “however” very often. Please use other terms for variety.

L95: This sentence is not easy to understand and should therefore be modified (e.g. studies are mainly conducted in other countries e.g. xxx; husbandry and management are not comparable to those in the USA)

L95: This aspect should be further explained. Why is it useful that the different environments are compared in a standardized approach?

L103: Why did you hypothesize, that free range eggs are superior? You have shown before, that literature is presenting conflicting results. I would try to formulate the research question in a more open-ended way.

L107: What is the 40th layer performance test? Please provide some more information.

L112: “Lohmann Tierzucht” instead of „Lohmann Tierzuckt“

L112: Please insert a blank space between value and unit (cm). Please harmonize within the whole manuscript.

L113: What was the third strain?

L114: The abbreviations have not been introduced before. Please use the written words (NCLP&MT).

L124: “cm²” instead of “cm2”

L139: Why did you use enrichable cages, if they are not used in practice?

L155: “C°” instead of “C”? Please harmonize within the manuscript.

Table 1: Please insert a blanket space between “<” and “70%”.

Table 2: Font and font size are partly different. Please standardize the table content.

Table 2: Please use two decimal places each in the table and in the text.

L182: Please insert the footnotes (1, 2, 3) in the table.

L182: “1,000” instead of “1000”? Please harmonize within the manuscript.

L191: Please explain why you chose these time points for data collection.

L192: The section on statistical methods should be the end of the material and methods chapter.

L194: Maybe better “fixed effects” instead of “main effects”?

L197: Why didn't you include genetic effects as a factor in the model? The significant effects suggest that genetics can affect egg quality. Therefore, you cannot generalize the results to all brown-laying hens because there could be genetic differences. This aspect should be discussed intensively.

L201: Who performed the egg quality measurements? What education or training did the person receive? Were there multiple assessors and was their assessor agreement tested?

L204: Why were these particular egg quality parameters selected? Please provide reasons for the selection of the different tests.

L216: Explain why you collected the data only at the times you did.

L220: Maybe “dry matter” instead of “dry mater”?

L228: Please delete the short sentence and insert (Table 3) in the previous sentence.

Table 3: A,B,C,D should be formatted as superscripts. Please standardize two decimal places (except p-values). This also applies to the other tables in the manuscript.

L282: Please standardize two decimal places also in the text.

Table 10: Please modify the table in a way that it fits on one page.

L344: The text for table 12 is too short in my opinion. Could you please describe the values in more detail?

L349: The manuscript contains relatively many small tables. Couldn't some tables be combined to save space? Due to the uniform structure, this should be possible without any problems.

L352: Please delete: “beginning with shell color”.

L353: Which shell color is preferred in the USA?

L361: Please correct: Dedousi et al. found…

L361: You have used the word “found” very often. Please use other terms for variety.

L364: Please insert a full stop after “et al.”

L376: Please insert the reference “Mills et al.” after this sentence.

L391: Please insert a full stop after “et al.” (2 times)

L392: Please insert a full stop after “et al.” (2 times)

L394: Dedousi et al.; Samiullah et al.

L395: Please insert a comma between CF and FR.

L395: Why did you mention here the prename?

L397: Please insert a full stop after “et al.”

L398: Please insert a full stop after “et al.”

L400: Please insert a full stop after “et al.”

L403: The sentence is too long. Please split into two sentences.

L417: Why did you mention here the prename?

L417: Maybe “and” instead of “end”?

L453: Please insert a comma after “membrane strength”.

L472: Your results do not allow such a conclusion. The offer of a green outdoor area gives the animals more possibilities than only the feed intake. Climatic stimuli, sunlight, space availability and exercise could also affect egg quality. Therefore, please delete the last sentence or formulate it as a need for further research for farms that cannot offer an outdoor run.

L472: As in the simple summary, I would write one more sentence here on the practical implications. Why is this study important for the farms?

L474: Please delete the first two sentences in this section. Please use the abbreviations mentioned at the front page of this manuscript (B.N.A. etc.)

L 495: The reference list contains many non-scientific publications. Can the information contained in it be found in scientific publications or government reports?

L498: Please provide a link for reference 2.

L534: Please correct the reference: Dikmen, B.Y.; Ipek, A;…

L536: The year is missing in this reference.

L543: Please delete “1” between “”Floor Pens” and “Poultr. Sci.”.

L545: “Geflügelkunde” instead of “Geflugelkunde“

L545: Please harmonize within the manuscript: “151-157” instead of “151–157”.

L555: Why “et al.”? Please mention all authors of this article.

L561: Please harmonize within the manuscript: “151-157” instead of “pp. 151-157”.

L583: The names of the authors are not correct. Please improve this citation.

L593: Please delete 14 between “Grashorn, M.” and “Feed additives”.

L595: The year should be written in bold letters.

L602: The year should be written in bold letters.

Author Response

Hello reviewer 2,

Thank you for your very thorough edit of my manuscript. As I have been asked to address each comment individual I will list the comment and my response below.

L10: Please also provide data from other states e.g. from Europe.

I have added a little information about Europe however, I wanted the manuscript to more focus on the US production as that is where a lot of the pressures are being applied to move towards extensive environments.

L13: The connection to egg quality is not clear from the explanations. The beginning of the sentence with "Therefore" does not fit. Please rephrase this section.

Thank you for letting me know this wasn’t clear. I have attempted to clarify.

L15: Please delete the first “and” in the brackets.

Deleted

L25: Why are only brown egg layers considered? Are significantly different results to be expected for white egg layers? What is the underlying hypothesis?

Brown egg layers were only considered for a few reasons. Firstly, while we performed research with white egg layers this was a separate study and was written about in a different manuscript. Secondly, I did not want to compare egg quality of brown and white egg layers. I was primarily concerned with how housing environment affected egg quality.

L26: In my opinion it is not necessary to divide the abstract in smaller sections. Please delete the subheadings (methods, results, conclusion).

Thank you for the clarification. I realize now that the instructions say not to include these so I have removed them.

L26: Please justify the hypothesis why egg quality should improve in the free range environment.

L28: The abbreviations have not been introduced before. Please use the written words (UFP, NRC, HSD).

L36: As in the simple summary, I would write one more sentence here on the practical implications. Why is this study important for the farms?

Thank you for the suggestions on the abstract. I have rewritten the abstract based on your suggestions as well as the suggestions of other reviewers

L32: You have used the word “found” very often. Please use other terms for variety.

I have gone through the manuscript and replaced found with other words.

L38: Please delete the semicolon at the end of the keywords.

Deleted

L44: I would refer here not only to consumer preferences, but also to industry requirements.

I have added a few words about what the industry prefers. Thank you for the suggestion

L54: Please give a reference for the statements concerning egg quality. 

L54: Please add another sentence as a transition between the sections: e.g. Egg quality can be significantly influenced by the housing environment (add reference)

Added some citations about consumer preference in yolk and shell color also added a sentence at the end as per suggestion. Also, I noticed that you asked me to delete the same sentence in the next paragraph. Should I include the added one here or also delete it?

L55: What is the situation in other countries? Please insert information for e.g. Europe.

Added a sentence about Europe.

L68: What is meant by the term "cage-free"? Please define this term.

I have deleted the cage free term here as I define it in more detail later

L70: This sentence is too long and complicated. Please split the sentence in two parts.

I have split this sentence into two parts.

L73: This sentence is a repetition and should therefore be deleted.

Thank you for your insight about this however, I would argue that this sentence is not repetitive this is the first time I have not discussed consumer preference for housing environment or willingness to pay yet. In what way is this sentence repetitive?

L78: This sentence appears again below in a similar form and should therefore be deleted here.

I have removed this sentence

L78: Please mention some influencing factors in this section. What could be the reasons for better or worse egg qualities (e.g. movement, light, nutrition...)?

Thank you for the suggestion. I have added some influencing factors which I think tie in nicely with the conclusion now.

L85: You have used the word “however” very often. Please use other terms for variety.

I have either deleted some “however” or changed for different words

L95: This sentence is not easy to understand and should therefore be modified (e.g. studies are mainly conducted in other countries e.g. xxx; husbandry and management are not comparable to those in the USA)

Thank you for pointing out when my writing may be confusing. I have attempted to change this sentence so it is easier to understand.

L95: This aspect should be further explained. Why is it useful that the different environments are compared in a standardized approach?

The importance to compare all of these environments in one standardized approach is to prevent any sort of confounding factors such as differences in ambient temperature and weather due to different seasons or nutritional and management practices between farms and regions.. I highlighted this in this sentence.

L103: Why did you hypothesize, that free range eggs are superior? You have shown before, that literature is presenting conflicting results. I would try to formulate the research question in a more open-ended way.

I added a sentence to explain why I chose my hypothesis. Specifically that while the research is conflicting it seems that free range hens have an advantage of better egg quality. Another review had commented that the hypothesis was not specific enough.

L107: What is the 40th layer performance test? Please provide some more information.

Information on the 40th layer performance test can be found in the citation

L112: “Lohmann Tierzucht” instead of „Lohmann Tierzuckt“

Changed. Thank you for the correction

L112: Please insert a blank space between value and unit (cm). Please harmonize within the whole manuscript.

I have included a blank space for every time I used a unit

L113: What was the third strain?

This was a typo. Thanks for catching

L114: The abbreviations have not been introduced before. Please use the written words (NCLP&MT).

I have included the abbreviation next to the first time I introduced it 

L124: “cm²” instead of “cm2”

Fixed

L139: Why did you use enrichable cages, if they are not used in practice?

In the US both enriched and enrichable colony cages are still used in the industry. I think the confusion may have come from the last sentence which I have removed.

L155: “C°” instead of “C”? Please harmonize within the manuscript.

Added the ° sign

Table 1: Please insert a blanket space between “<” and “70%”.

Space added

Table 2: Font and font size are partly different. Please standardize the table content.

Thank you for the catch. I have standardized the font.

Table 2: Please use two decimal places each in the table and in the text.

I used more than two decimal places in the cases where the decimal starts at the hundredths place in order to be more accurate. If this is a major issue I can still change these although I feel some accuracy is lost if I must round to the hundredths place. We believe when we publish feed formulas that it is important to have the exact formulas so this study can be replicated.

L182: Please insert the footnotes (1, 2, 3) in the table.

Thank you for the catch. Footnotes inserted

L182: “1,000” instead of “1000”? Please harmonize within the manuscript.

I have added the “,” for numbers at or over a thousand

L191: Please explain why you chose these time points for data collection.

The time points used were to represent Pre-peak, peak, post peak, mid cycle, late cycle and end cycle. The time points for the egg solids were to represent peak production, mid cycle and late cycle

L192: The section on statistical methods should be the end of the material and methods chapter.

Thanks for the suggestion. I have moved this section

L194: Maybe better “fixed effects” instead of “main effects”?

Thank you for the suggestion. I used main effects because I did not use any variable effects and therefore all the main effects are fixed effects. Other papers from Animals used the term main effects so that is why I chose that term.

L197: Why didn't you include genetic effects as a factor in the model? The significant effects suggest that genetics can affect egg quality. Therefore, you cannot generalize the results to all brown-laying hens because there could be genetic differences. This aspect should be discussed intensively.

Thank you for the comment. The point of the research was to explore how the average brown egg layer would respond, which is why we used essentially the average of two strains in the model. Due to the nature of the research, we are unable to discuss strain differences in the manuscript. Moreover, as the p-values of the statistical analysis are incredibly low for housing environment, adding genetic effect would not change the results we found and would furthermore, cause a drastic loss of degrees of freedom in the statistical model resulting in a weaker model. Also, I did run a model that included genetic effect and the AIC was better for the model that did not include the genetic effect.

L201: Who performed the egg quality measurements? What education or training did the person receive? Were there multiple assessors and was their assessor agreement tested?

Added a statement on trained laboratory personnel

L204: Why were these particular egg quality parameters selected? Please provide reasons for the selection of the different tests.

These egg quality measurements are the standard measurements performed in out lab that have been informed by past literature as standard egg quality measurements.

L216: Explain why you collected the data only at the times you did.

The time points used were to represent Pre-peak, peak, post peak, mid cycle, late cycle and end cycle. The time points for the egg solids were to represent peak production, mid cycle and late cycle

L220: Maybe “dry matter” instead of “dry mater”?

Fixed

L228: Please delete the short sentence and insert (Table 3) in the previous sentence.

Deleted

Table 3: A,B,C,D should be formatted as superscripts. Please standardize two decimal places (except p-values). This also applies to the other tables in the manuscript.

All multiple comparisons have been formatted as superscripts

L282: Please standardize two decimal places also in the text.

I have standardized all data to reflect at most two decimals except for P values, some SEMs and for vitelline membrane strength and shell elasticity as I feel that removing the third decimal will result in a loss of accuracy. For example comparing cage free and free range at week 35 for vitelline membrane strength. Both the numbers would show 0.02 if only two decimals were used however the multiple comparisons are different, A in the case of free range and AB in the case of cage free

Table 10: Please modify the table in a way that it fits on one page.

It is difficult to ensure that all tables are on a singular page until I know what edits are accepted and which are not. This may need to be the final thing I do to ensure consistency although I have made an attempt.

L344: The text for table 12 is too short in my opinion. Could you please describe the values in more detail?

I added some descriptive language however it is difficult to extend this section without adding fluff as we did not see any differences.

L349: The manuscript contains relatively many small tables. Couldn't some tables be combined to save space? Due to the uniform structure, this should be possible without any problems.

Thank you for the suggestion. However, We feel like while there are several tables, keeping these tables separate allows the reader easier access to find the information they may be looking for.

L352: Please delete: “beginning with shell color”.

Deleted

L353: Which shell color is preferred in the USA?

Thank you for this question! Darker shells are preferable. Sentence added with reference to support this

L361: Please correct: Dedousi et al. found…

Corrected

L361: You have used the word “found” very often. Please use other terms for variety.

I have added more variety

L364: Please insert a full stop after “et al.”

Inserted a full stop

L376: Please insert the reference “Mills et al.” after this sentence.

Inserted the reference

L391: Please insert a full stop after “et al.” (2 times)

Inserted a full stop

L392: Please insert a full stop after “et al.” (2 times)

Inserted a full stop

L394: Dedousi et al.; Samiullah et al.

Fixed the grammar error

L395: Please insert a comma between CF and FR.

Comma inserted

L395: Why did you mention here the prename?

I thought that the prename was part of the last name. This has been fixed.

L397: Please insert a full stop after “et al.”

Inserted a full stop

L398: Please insert a full stop after “et al.”

Inserted a full stop

L400: Please insert a full stop after “et al.”

Inserted a full stop

L403: The sentence is too long. Please split into two sentences.

Broke up the sentences

L417: Why did you mention here the prename?

I thought that the prename was part of the last name. This has been fixed.

L417: Maybe “and” instead of “end”?

Fixed the typo

L453: Please insert a comma after “membrane strength”.

Added the comma

L472: Your results do not allow such a conclusion. The offer of a green outdoor area gives the animals more possibilities than only the feed intake. Climatic stimuli, sunlight, space availability and exercise could also affect egg quality. Therefore, please delete the last sentence or formulate it as a need for further research for farms that cannot offer an outdoor run.

 L472: As in the simple summary, I would write one more sentence here on the practical implications. Why is this study important for the farms?

I added a few sentences at the end discussing the effects to the producers as well as rewrote the other sentence to discuss that further research is required.

L474: Please delete the first two sentences in this section. Please use the abbreviations mentioned at the front page of this manuscript (B.N.A. etc.)

Replaced full names with initials

L 495: The reference list contains many non-scientific publications. Can the information contained in it be found in scientific publications or government reports?

We felt that while these references are not government or scientific publications, I still needed to reference them as they are the announcements in popular press about the switch from cage to cage free. We did not use any of these references to compare the research we did to.

L498: Please provide a link for reference 2.

Link provided

L534: Please correct the reference: Dikmen, B.Y.; Ipek, A;…

Corrected the naming

L536: The year is missing in this reference.

Added the tear

L543: Please delete “1” between “”Floor Pens” and “Poultr. Sci.”.

Deleted the 1

L545: “Geflügelkunde” instead of “Geflugelkunde“

 Thank you for catching the misspelled word here

L545: Please harmonize within the manuscript: “151-157” instead of “151–157”.

Unsure why my citation manager did this. I used the zotero package that was linked by MDPI. All dashes have been shortened

L555: Why “et al.”? Please mention all authors of this article.

Zotero must have shortened this entry as there were many authors. I have included them all by name.

L561: Please harmonize within the manuscript: “151-157” instead of “pp. 151-157”.

Same as above

L583: The names of the authors are not correct. Please improve this citation.

Corrected the names. Thanks for the catch!

L593: Please delete 14 between “Grashorn, M.” and “Feed additives”.

Deleted

L595: The year should be written in bold letters.

Bolded the year

L602: The year should be written in bold letters.

Bolded the year

Thank you for taking the time to help me make this manuscript better. I sincerely appreciate the level of detail contained within your review.

Please let me know if I can do anything else,

Benjamin Alig, Ramon Malheiros, Kenneth Anderson 

Reviewer 3 Report

Although a lot of work went into this manuscript, there are areas that must be improved. It is impressive that the researchers were able to compare five different housing environments on physical egg quality.  However, in its' current form, it lacks substance in terms of what these differences in physical egg quality of commercial brown egg layers mean.  Why do these different environments impact these characteristics? There is no connection between the egg quality and the hen.  Primarily focused on consumer perception.  The introduction does not provide enough information to support the objectives and hypothesis (both are vague).  The results are cumbersome and the discussion is not informative.  

General comments:

The title is vague and lacks substance.  Also, after reading the title one expects that the results will tell us what and why these different environments affect egg quality and what that means. 

The abstract also lacks conciseness and is not informative, especially in terms of supporting the conclusion. 

The overall conclusion states free range environment resulted in superior quality eggs compared to all other environments and enrichment does not affect the quality of eggs.  However, not sure what superior means and which data support this strong conclusion.  Why is the quality of the egg so important -- does it connect with hen welfare and or human nutrition? 

Introduction--1st paragraph does not set the stage per se.  It seems to be independent of the next two paragraphs.  Your introduction does not set the stage for your vague objectives or your hypothesis.  

There is no discussion on what factors or why the housing environment has an impact on physical eqq quality parameters. The discussion is mostly repeating the results and using data that agrees or disagrees with their data. Not very informative.   

Author Response

Hello reviewer 3,

Thank you for your comments on my manuscript. You have helped me reshape some small parts of the manuscript to allow for a more directional paper. As I have been asked to address each comment individual I will list the comment and my response below.

The title is vague and lacks substance.  Also, after reading the title one expects that the results will tell us what and why these different environments affect egg quality and what that means.

I have changed the title to be more descriptive and specific. Hanks for this suggestion!

The abstract also lacks conciseness and is not informative, especially in terms of supporting the conclusion.

I have completely rewritten the abstract as other reviewers have asked me to.

The overall conclusion states free range environment resulted in superior quality eggs compared to all other environments and enrichment does not affect the quality of eggs.  However, not sure what superior means and which data support this strong conclusion.  Why is the quality of the egg so important -- does it connect with hen welfare and or human nutrition?

From the measurements we performed. These quality parameters are most important for the production and marketing side of the egg industry. While it is possible that a change in quality may relate to nutrition, we are unable to draw these conclusions without further research. For welfare, I discussed in the discussion that elevated levels of stress may be a cause for differences that are seen however we did not specifically measure stress and are therefore unable to draw that conclusion. I have also added a few sentences per another reviewers request about how moving towards extensive environments

Introduction--1st paragraph does not set the stage per se.  It seems to be independent of the next two paragraphs.  Your introduction does not set the stage for your vague objectives or your hypothesis. 

Thank you for the suggestion. I think that both paragraphs are important to have however, after some thought I have switched the first and second paragraph because I believe you are right in that that the original first paragraph does not set the hook in the reader like the second paragraph could.

There is no discussion on what factors or why the housing environment has an impact on physical eqq quality parameters. The discussion is mostly repeating the results and using data that agrees or disagrees with their data. Not very informative. 

Unfortunately, due to the nature of the research, we were unable to tackle why housing environment caused differences in egg quality. We have presented several theoreticals and supporting research for them however, as we did not perform any further tests this question will need to be left up to further research. The point of this research was to simply evaluate egg quality under several different housing environments, which has been made more clear with your suggestion of a title revision.

Thank you again for your suggestions. Yours in particular have helped me attempt to reshape some of the manuscript to be clearer in what we were trying to tackle.

Please let me know if there is anything else I can do,

Benjamin Alig, Ramon Malheiros, Kenneth Anderson

Reviewer 4 Report

General comments:

The manuscript “The effect of housing environment on physical egg quality of 2 commercial brown egg layers” is an important and actual topic. The research increases the knowledge of egg quality in different housing systems. The paper fits well with the scope of the journal. However, in my opinion, it has many shortcomings in this current form. In particular, my concerns are:

1.      I suggest rewriting the abstract to include more results observed

2.      According to the instructions for authors, the abstract should be a single paragraph without headings

3.      I suggest including a sentence in the Introduction on the relationship between housing systems and egg defects; for example, you can read and cite 10.3390/ani12182307

4.      I suggest rewriting the materials and methods section to make it clearer. May be useful to divide it into subsections.

5.      I also recommend that the methods of analysis be better specified, including citation of the references from which the equations were taken

Specific comments:

L26. Delete ‘Methods:’.

L31: Delete ‘Results:’

L34: Delete ‘Conclusion:’

L37-38: In general, I’d go for words that are not included in the title. For example, I suggest using: Egg weight; Haugh Unit; Shell color; Enrichments;

LL42-45: Please report citation, for example, you can read and cite 10.1016/j.psj.2020.06.064

LL79-80: Please report the studies.

LL86-89: Please rewrite the sentence to make it clearer.

LL109-110: It is not necessary to report the aim of the study in the M&M sections, please delete.

LL110-113: In this sentence, you stated that there were 3 strains used, but you reported only the names of two. How many strains did you use? Also, state the number of animals in each strain.

LL111-113: Please include the total number of hens involved in the research.

LL115-117: I assume that the number of hens was the same for the different housing systems, however, I suggest specifying it.

LL174-175: Are you sure? In the text, Table 3 is stated to report the shell color... What is the correct information?

LL180-181: Please check the title and include the correct table number.

L225: Please be consistent with the results section, if you decide to create several subsections do it for all the results.

LL355-356: please check this sentence and use the correct abbreviation.

L357 and elsewhere: Please report the number corresponding to the citation after the author's name.

L361: Dedousi et al. [41] found.

Author Response

Hello Reviewer 4,

Thank you for your suggestion and your opening comments about the validity of the research. Hopefully I have met your concerns with my edits. As I have been asked to address each comment individual I will list the comment and my response below

  1. I suggest rewriting the abstract to include more results observed

I have completely rewritten the abstract

  1. According to the instructions for authors, the abstract should be a single paragraph without headings

Thank you for the catch. I see now where the instructions say this. The headings have been removed.

  1. I suggest including a sentence in the Introduction on the relationship between housing systems and egg defects; for example, you can read and cite 10.3390/ani12182307

I have added this citation. Thank you for the suggestion!

  1. I suggest rewriting the materials and methods section to make it clearer. May be useful to divide it into subsections.

I have divided the materials and methods into subsections. Thank you for suggesting that.

  1. I also recommend that the methods of analysis be better specified, including citation of the references from which the equations were taken

I have cited the Haugh unit equation however, I did not think that dry egg matter needed to be cited as it is a simple formula.

Specific comments:

L26. Delete ‘Methods:’.

Deleted

L31: Delete ‘Results:’

Deleted

L34: Delete ‘Conclusion:’

Deleted

L37-38: In general, I’d go for words that are not included in the title. For example, I suggest using: Egg weight; Haugh Unit; Shell color; Enrichments;

Thank you for the suggestion. We think that it is best to keep the keywords in as they couldn’t hurt a search function

LL42-45: Please report citation, for example, you can read and cite 10.1016/j.psj.2020.06.064

Thank you for the suggestion. I have added a citation.

LL79-80: Please report the studies.

I removed the first sentence as I cite many past studies in the body of that paragraph so the sentence was not needed

LL86-89: Please rewrite the sentence to make it clearer.

Removed the first part of the sentence to make the sentence clearer and moved the citation from the middle of the sentence to the end.

LL109-110: It is not necessary to report the aim of the study in the M&M sections, please delete.

Removed the aim of the study from the M&M

LL110-113: In this sentence, you stated that there were 3 strains used, but you reported only the names of two. How many strains did you use? Also, state the number of animals in each strain.

Corrected the typo. There were two strains involved.

LL111-113: Please include the total number of hens involved in the research.

Included total number of hens in each strain and total number of hens involved

LL115-117: I assume that the number of hens was the same for the different housing systems, however, I suggest specifying it.

Yes the same number of hens were used the previous fix should clarify this. Thank you for pointing this out.

LL174-175: Are you sure? In the text, Table 3 is stated to report the shell color... What is the correct information?

Yes I had a typo in the table numbers here. These should be fixed now

LL180-181: Please check the title and include the correct table number.

Table numbers are correct now

L225: Please be consistent with the results section, if you decide to create several subsections do it for all the results.

Subsections that I missed are inserted now

LL355-356: please check this sentence and use the correct abbreviation.

Changed “percentage” to “%”

L357 and elsewhere: Please report the number corresponding to the citation after the author's name.

L361: Dedousi et al. [41] found.

In the discussion, I moved the citations to be in front of the first authors if I had them.

Thank you again for your comments. I hope I have addressed them accurately.

Please let me know if there is anything else I can do,

Benjamin Alig, Ramon Malheiros, Kenneth Anderson

Round 2

Reviewer 2 Report

General comments:

The manuscript animals-2178064, entitled "Evaluation of physical egg quality parameters of commercial brown laying hens housed in five production systems " with Mr. Alig as first author was properly revised by the authors. The reviewer comments were considered in the revision and the authors left none of my questions unanswered. However, some minor inconsistencies and logical mistakes can still be found in the manuscript. Various comments for suggested changes to the text could be find in the following section (Detailed comments).

Detailed comments:

 L10: Please delete the comma between the two sentences (…years., Recently…)

L23: Why are only brown egg layers considered? Are significantly different results to be expected for white egg layers? What is the underlying hypothesis?

Brown egg layers were only considered for a few reasons. Firstly, while we performed research with white egg layers this was a separate study and was written about in a different manuscript. Secondly, I did not want to compare egg quality of brown and white egg layers. I was primarily concerned with how housing environment affected egg quality.

I am not convinced by this argumentation. If you expect differences between different genetics, a comparison between white and brown laying hens would be very interesting. If there is probably no difference between the genetics, you could look at the results together and include genetic background as a factor in the analysis. In my view, the publication of two almost identical manuscripts is not permissible. Therefore, you should think again about the combination of the two manuscripts.

L35: As in the simple summary, I would write one more sentence here on the practical implications. Why is this study important for the farms? Please add the re-phrased sentence from the simple summary: “This information shows that…”

L58: Please insert a blanket space between “cages” and the bracket.

L60: Could you replace one of the two “although” by another term?

L62: This sentence is a repetition and should therefore be deleted.

Thank you for your insight about this however, I would argue that this sentence is not repetitive this is the first time I have not discussed consumer preference for housing environment or willingness to pay yet. In what way is this sentence repetitive?

In my opinion it is not necessary to repeat the information concerning legislators, groceries and food services (see L55). It is certainly enough to mention the trend towards cage-free housing.

L90: Please insert a blanket space between “shells” and the bracket.

L91: Please harmonize within the manuscript: “egg shells” or “eggshells”.

L116: Please insert a blanket space between “Tierzucht” and “GmbH”.

L117: Maybe “1,648 hens” instead of “1648 hens”?

L128: Maybe “6,131.6 cm²” instead of “6131.6 cm²”?

L149: Maybe “1,141 cm²” instead of “1141 cm²”?

L150: The abbreviation UEP was not introduced before.

L152: Maybe “1,233 cm²” instead of “1233 cm²”?

L153: Maybe “1,141 cm²” instead of “1141 cm²”?

L154: Please insert a blanket space between “16” and “cm”.

L169: Maybe “1,141 cm²” instead of “1141 cm²”?

L179: The abbreviation NRC was not introduced before.

Table 1: Please harmonize within the table: blanket space between value and unit?

L187: Please correct: “…described in Table 1.

Table 2: Perhaps you could always use the same number of decimals for uniformity? For example 4.00 % Calcium instead of 4 %?

L274: Please insert a blanket space between “1” and “mm”.

L331: Please insert a blanket space between “0.087” and “mm” (applies to the whole paragraph).

L402: Please insert a blanket space between “full stop” and “Further”.

L419: Please correct: “identified” instead of “Identified”.

L491: Recognizing that environmental enrichment does not automatically improve egg quality is very important. Could you go into this aspect in a little more detail in the discussion? How do you justify these unexpected results?

L504: Perhaps mention at this point that this could also bring economic benefits and at least partially compensate for the additional costs?

L515: Please add a full stop at the end of the sentence.

Author Response

Hello reviewer 2,

Thank you for going through and reviewing my manuscript again. Hopefully you will find these edits to your satisfaction.

L10: Please delete the comma between the two sentences (…years., Recently…)

Deleted the comma and the extra space that was there

(Reviewer 2's original comment) L23: Why are only brown egg layers considered? Are significantly different results to be expected for white egg layers? What is the underlying hypothesis?

(Mr. Alig and other author's original response): Brown egg layers were only considered for a few reasons. Firstly, while we performed research with white egg layers this was a separate study and was written about in a different manuscript. Secondly, I did not want to compare egg quality of brown and white egg layers. I was primarily concerned with how housing environment affected egg quality.

(Reviewer 2's round second comment): I am not convinced by this argumentation. If you expect differences between different genetics, a comparison between white and brown laying hens would be very interesting. If there is probably no difference between the genetics, you could look at the results together and include genetic background as a factor in the analysis. In my view, the publication of two almost identical manuscripts is not permissible. Therefore, you should think again about the combination of the two manuscripts.

(Mr. Alig and other author's response to the second comment): While I absolutely agree with you that it would be incredibly interesting for a larger comparison between whites and brown egg layers in housing environments I left the manuscripts separate for a few reasons. Firstly, we did not have white egg layers in the same environments as brown egg layers. Specifically, we did not have any replications of free range for white egg layers. Therefore, statistically, it would have been much more difficult and convoluted to compare the genetic strains. Secondly, some of the methods used were different between brown and white egg layers (such as replications, stocking density as well as parameters measured and dates these parameters were measured) and therefore I did not feel comfortable comparing these two as I would be worried about confounding factors due to these.

L35: As in the simple summary, I would write one more sentence here on the practical implications. Why is this study important for the farms? Please add the re-phrased sentence from the simple summary: “This information shows that…”

Added a sentence at the end of the abstract about this.

L58: Please insert a blanket space between “cages” and the bracket.

Added a blank space

L60: Could you replace one of the two “although” by another term?

Removed one and replaced the other one with however

(Reviewer 2's original comment) L62: This sentence is a repetition and should therefore be deleted.

(Mr. Alig and other author's original comment): Thank you for your insight about this however, I would argue that this sentence is not repetitive this is the first time I have not discussed consumer preference for housing environment or willingness to pay yet. In what way is this sentence repetitive?

(Reviewer 2's second comment): In my opinion it is not necessary to repeat the information concerning legislators, groceries and food services (see L55). It is certainly enough to mention the trend towards cage-free housing.

(Mr. Alig and other author's response to reviewer 2's second comment): understand what you are referring to. Thank you for clearing that up. I kept the sentence but I deleted the part rehashing the info about legislators, grocers and food services. I wanted to indicate that while consumers aren’t willing to switch if given a choice, they may not actually have a choice.

L90: Please insert a blanket space between “shells” and the bracket.

Space inserted

L91: Please harmonize within the manuscript: “egg shells” or “eggshells”.

Changed al to eggshells.

L116: Please insert a blanket space between “Tierzucht” and “GmbH”.

Space inserted

L117: Maybe “1,648 hens” instead of “1648 hens”?

Comma added. Thank you for pointing these out. I apologize for not catching these last round.

L128: Maybe “6,131.6 cm²” instead of “6131.6 cm²”?

Comma added

L149: Maybe “1,141 cm²” instead of “1141 cm²”?

Comma added

L150: The abbreviation UEP was not introduced before.

Removed the acronym and put the full name

L152: Maybe “1,233 cm²” instead of “1233 cm²”?

Comma added

L153: Maybe “1,141 cm²” instead of “1141 cm²”?

Comma added

L154: Please insert a blanket space between “16” and “cm”.

Space added

L169: Maybe “1,141 cm²” instead of “1141 cm²”?

Comma added

L179: The abbreviation NRC was not introduced before.

Abbreviation removed and full name inserted

Table 1: Please harmonize within the table: blanket space between value and unit?

Removed the space between 90 and % in order to be consistent

L187: Please correct: “…described in Table 1.

Fixed. Thank you for pointing this out.

Table 2: Perhaps you could always use the same number of decimals for uniformity? For example 4.00 % Calcium instead of 4 %?

Added the extra decimal places

L274: Please insert a blanket space between “1” and “mm”.

Added the space

L331: Please insert a blanket space between “0.087” and “mm” (applies to the whole paragraph).

Added a space to all instances

L402: Please insert a blanket space between “full stop” and “Further”.

Space added

L419: Please correct: “identified” instead of “Identified”.

Word corrected

L491: Recognizing that environmental enrichment does not automatically improve egg quality is very important. Could you go into this aspect in a little more detail in the discussion? How do you justify these unexpected results?

This is a great point. Thank you for saying this. I have some words that the biggest improvement in egg quality seemed to be when hens were removed from the cages and identified several reasons specific to cage free and free range that could have possibly caused the improvement in egg quality.

L504: Perhaps mention at this point that this could also bring economic benefits and at least partially compensate for the additional costs?

Added a sentence about this before the final conclusions at the end.

L515: Please add a full stop at the end of the sentence.

Period added.

Reviewer 4 Report

The authors have satisfactorily responded to all my questions and made the necessary changes to the manuscript. The revised version of the manuscript appears to be good. It looks ready for publication as far as I can tell.

Author Response

Thank you for your kind words and for helping me refine and perfect this manuscript. I am glad for the help!